# A New Method for Heart Disease Detection: Long Short-Term Feature Extraction from Heart Sound Data

**DOI:** 10.3390/s23135835

**Published:** 2023-06-23

**Authors:** Mesut Guven, Fatih Uysal

**Affiliations:** 1Gendarmerie and Coast Guard Academy, Ankara 06805, Turkey; 2Department of Electrical and Electronics Engineering, Faculty of Engineering and Architecture, Kafkas University, Kars 36100, Turkey; fatih.uysal@kafkas.edu.tr

**Keywords:** machine learning, long short-term features, feature selection, auscultation, heart abnormalities, heart sound classification

## Abstract

Heart sounds have been extensively studied for heart disease diagnosis for several decades. Traditional machine learning algorithms applied in the literature have typically partitioned heart sounds into small windows and employed feature extraction methods to classify samples. However, as there is no optimal window length that can effectively represent the entire signal, windows may not provide a sufficient representation of the underlying data. To address this issue, this study proposes a novel approach that integrates window-based features with features extracted from the entire signal, thereby improving the overall accuracy of traditional machine learning algorithms. Specifically, feature extraction is carried out using two different time scales. Short-term features are computed from five-second fragments of heart sound instances, whereas long-term features are extracted from the entire signal. The long-term features are combined with the short-term features to create a feature pool known as long short-term features, which is then employed for classification. To evaluate the performance of the proposed method, various traditional machine learning algorithms with various models are applied to the PhysioNet/CinC Challenge 2016 dataset, which is a collection of diverse heart sound data. The experimental results demonstrate that the proposed feature extraction approach increases the accuracy of heart disease diagnosis by nearly 10%.

## 1. Introduction

Heart disease is the main cause of death globally [1]. Thus, detecting heart defects through artificial intelligence and machine learning algorithms will positively affect health services globally [2]. One of the methods used for detecting heart diseases is auscultation via stethoscopes. Heart rhythm disorders, heart valve diseases, and congenital heart diseases can be detected through auscultation. The stethoscopes used for auscultation have transformed into devices with the capability to record and filter heart sounds [3,4,5]. Heart sounds can now be uploaded to a computer, tablet, or smartphone and can be shared through e-mails and messaging applications thanks to the significant capabilities of modern digital stethoscopes. In this work, the PhysioNet/CinC Challenge 2016 dataset, which consists of 2435 recordings, is used [6]. To meet the sample size requirements, in most heart sound classification algorithms, heart sound instances are divided into small fragments [7,8]. Afterward, features are extracted from each fragment for classification. This method lacks information about the neighboring fragments and the heart sound instance to which the corresponding fragment belongs for classification. To compensate for this drawback, we propose two different periods named short-term and long-term periods for extracting features. Short-term features are extracted from each window and are specific to the corresponding fragment, whereas long-term features are extracted from the heart sound instances to which the fragments being classified belong. To create a more informative feature set, long-term features are combined with fragment-specific short-term features, resulting in a merged feature set known as a long short-term feature set. In the experiments, short-term features are extracted using both Mel Frequency Cepstral Coefficients and the statistical properties of the signals. Additionally, long-term features are extracted by using the duration properties of fundamental heart sounds. There are 27 short-term features and 6 long-term features. Thus, the dimension of each fragment in the long short-term feature set is 33. Among these 33 features, the most informative ones are determined through nearest component analysis. After the feature dimension reduction process, the selected long short-term feature set performs better than the selected short-term feature set. Additionally, the classification performance of these selected features is measured. The experiments involving the merging of long short-term features and nearest component analysis reveal not only an increase in the overall accuracy rate but also that the information carried by long-term features is much richer than that carried by short-term ones.

The motivation for our work and some of our contributions to the literature can be summarized as follows:Heart sound classification remains a prominent topic of discussion, and the PhysioNet/CinC Challenge 2016 dataset is currently the most comprehensive and up-to-date collection of data available. With the dataset being made available to the public, numerous machine learning algorithms have been applied to the data, most of which divide signals into fragments and then analyze the features extracted from those fragments. This study stands out as the only one representing the features extracted from both small fragments and the entire heart sound signal.The main contribution of this work lies in its assertion that small fragments and the whole signal have distinct characteristics, and when used in combination, they increase classification accuracy. To validate this assertion, we employ various machine learning models utilizing a combination of features. The results indicate that the combined feature set boosts the classification accuracy on the publicly available portion of the PhysioNet dataset.Lastly, we propose a novel approach to eliminate extraneous peaks and determine the fundamental heart sounds.

## 2. Related Works

Heart sound segmentation and classification studies have been conducted for a long time. To provide an up-to-date account, this section summarizes studies conducted within the last 10 years. In 2010, Schmidt et al. segmented fundamental heart sounds (FHSs) using a hidden semi-Markov model (HSMM). Their strategy was based on extracting features from systolic and diastolic durations. They achieved a 98.8% segmentation accuracy [9]. Ari et al. (2010) extracted wavelet features and used a modified version of the Support Vector Machine (SVM) algorithm for the classification. The modified SVM and wavelet features resulted in an 86.72% accuracy [10]. Avendano-Valencia et al. (2010) carried out experiments on 26 normal and 19 pathological heart sounds to identify the most informative features [11]. In 2011, Bentley et al. initiated a heart sound segmentation and classification challenge [12]. They also released a public heart sound database called the PASCAL database. Gharehbaghi et al. (2011) conducted a feature-based segmentation of FHSs on 120 child heart sound recordings and achieved a 97% segmentation accuracy [13]. Another important achievement in 2011 was the increase in the number of available heart sounds. For example, Li et al. (2011) contributed to The Dalian University of Technology’s heart sound database [14] and Moukadem et al. (2011) contributed to the University of Haute Alsace’s heart sound database [15]. In 2012, Tang et al. proposed a segmentation method that separated heart cycles in the frequency domain and then clustered the FHSs. They achieved a 95% segmentation accuracy [16]. Uguz implemented an artificial neural network that utilized wavelet features for analyzing three classes, normal, pulmonary stenosis, and mitral stenosis, and achieved an average accuracy of 98.33% for the three classes [17].

In 2013, Moukadem et al. proposed a method that utilized Shannon energy for segmenting FHSs. The segmentation accuracy of the method was 95% on 80 samples, with an equal number of samples from each class [18]. Naseri and Homaeinezhad (2013) utilized over 52 heart sounds for segmenting FHSs. They extracted features from both the frequency and time domains [19]. Castro et al. (2013) carried out segmentation on the PASCAL database, resulting in a 90.1% accuracy for first heart sound segmentation and a 93.3% accuracy for second heart sound segmentation [20]. In 2014, Sun et al. carried out segmentation on the Michigan Database using the Hilbert transform [21]. Varghees and Ramachandran (2014) used a feature-based method, along with Shannon entropy [22]. Pedrosa et al. (2014) used periodic component features for segmenting the FHSs and achieved a 98.6% accuracy on 72 heart sounds [23]. Papadaniil and Hadjileontiadis also implemented a feature-based segmentation method that detected the starting and ending points of FHSs [24]. In 2015, Zheng et al. utilized an SVM classifier that uses wavelet features on 107 heart sounds and achieved a 97.17% classification accuracy [25]. Patidar et al. used a modified and tunable version of wavelet transform as features for the SVM classifier, achieving a reported 98.8% sensitivity on 163 heart recordings [26]. Gharehbaghi et al. implemented a modified version of the SVM for two equally distributed classes of 60 heart sound samples, reporting an 86.4% classification accuracy [27]. In 2016, PhysioNet released the largest public heart sound database ever, which consists of seven different databases. The source and other properties of each database are defined in PyhsioNet’s well-known paper, along with the entire history of heart sound segmentation and classification works [6]. In the same year, a competition called the PhysioNet/Computing in Cardiology (CinC) Challenge 2016 was held. Participants implemented their segmentation and classification algorithms on the PhysioNet database. Springer et al. (2016) improved Schmidt et al’s work on the hidden semi-Markov model [28]. Their proposed method has achieved superior accuracy rates compared to the current methods and is currently considered the best method for segmenting heart sounds into FHSs. In 2018, Yu Tsaoa et al. recorded heart sounds in a noise-free environment and then added synthetic noise to the original noiseless heart sounds. resulting in two different types of data. They trained two deep learning algorithms using the noiseless heart sounds and a combination of noiseless data and noisy data. The authors reported that the second deep learning algorithm that was trained on the combined data outperformed the other algorithm in classifying noisy heart sounds [29]. Siddique Latif et al. implemented a recurrent neural network (RNN) on the PhysioNet 2016 database and compared the RNN with other deep learning models [8]. They reported achieving a 97.63% accuracy by dividing the available data into 75% for training, 15% for validation, and 10% for testing. The RNN outperformed the other deep learning algorithms. Juan et al. (2018) implemented a neural network-based classification on a field-programmable gate array. They used a modified version of the AlexNet model, achieving a 97% accuracy [30]. The use of Mel Frequency Cepstral Coefficients (MFFCs) for extracting some of the short-term features in this study is considered one of the most popular techniques for extracting features in digital voice recognition tasks. The MFFC technique was first introduced to replicate the human hearing mechanism [31]. From the time they were introduced, MFFCs have been widely used for extracting short-time-period features. Although MFFCs are useful, they need to be supplemented with other features. In the literature, especially in audio-signal processing for speech, there have been efforts to enhance MFFCs by incorporating long term-features that contain complementary information not detected in the short-time period alone [32].

## 3. Materials and Methods

### 3.1. Dataset

In all the experiments, the PhysioNet Computing in Cardiology 2016 database was used. This dataset is a combination of nine different databases formed by independent research groups at different locations and times. Since the dataset consists of different databases, the quality, recording length, and sampling frequency of the recordings are different. To equalize the sampling frequencies, all recordings are set to a sampling frequency of 2000 kHz. There are 2435 labeled heart sounds in this dataset, consisting of two classes: normal and abnormal. Normally labeled samples are used to represent healthy heart sounds, which only contain the first (S1) and second (S2) heart sounds. Abnormally labeled samples are used to represent unhealthy heart sounds from confirmed cardiac patients. Unhealthy heart sounds exhibit additional noisy patterns alongside the fundamental heart sounds. The first and second heart sounds are called fundamental heart sounds. A cardiac period begins with S1 and is followed by S2. The time interval between S1 and S2 is called the systolic period and the time interval between S2 and S1 is called the diastolic period, as shown in Figure 1.

As stated above, healthy heart sounds comprise only S1 and S2 patterns, which are produced from the contraction and relaxation movements of the heart. On the other hand, unhealthy heart sounds exhibit additional noisy patterns alongside S1 and S2. The difference between a healthy and an unhealthy heart sound is presented in Figure 2.

### 3.2. Proposed Method

Heart sound classification tasks consist of three main steps, as shown in Figure 3. These steps include pre-processing, feature extraction, and classification [33,34,35]. They are conducted on a small window of the examined heart sound [36].

The original contribution of this study is to extract features not only from a small window but also from the entire heart sound signal. The first-time scale is defined as the short-term scale and the second time scale is defined as the long-term scale. In the short-term scale, heart sounds are divided into fixed-sized small fragments, and short-term features are extracted from those fragments. In the long-term scale, heart sound instances are considered as a whole, and long-term features are extracted from each instance. Classification is carried out on each fragment. So, for each fragment, the short-term features extracted from the respective fragment and the long-term features extracted from the instance to which the respective fragment belongs are merged to create a mixed set of features. The feature-merging process enhances the overall representation thanks to the useful information carried by the long-term features.

The proposed method consists of four main steps and nine sub-steps. The main steps are pre-processing, feature extraction, feature pooling, and classification. To provide a clearer understanding, a flowchart of the proposed methodology is presented in Figure 4.

The proposed method, which aims to extract features not only from small window segments but also from the entire heart sound signal, represents an original contribution of this study. The method involves a two-time-scale approach, where the short-term time scale refers to fixed-sized fragments of heart sounds, and the long-term time scale encompasses complete heart sound instances. In the short-term time scale, the heart sounds are partitioned into small fragments of a predefined size, and short-term features are extracted from each fragment. On the other hand, in the long-term time scale, the heart sound instances are treated as a whole, and long-term features are extracted from each instance. The classification process is carried out on each fragment, resulting in a comprehensive analysis. To create a comprehensive set of features for each fragment, the short-term features extracted from the respective fragment are combined with the long-term features extracted from the instance to which the fragment belongs. This merging process enhances the overall representation by incorporating the valuable information carried by the long-term features. The proposed method is outlined in detail through four key steps, namely preprocessing, feature extraction, feature pooling, and classification. The preprocessing step involves necessary data cleaning and noise reduction techniques to ensure the quality of the heart sound signals. Next, the feature extraction step focuses on extracting relevant features from both short-term fragments and long-term instances. Subsequently, the feature pooling step combines the short-term and long-term features to create a merged feature set that captures the comprehensive characteristics of the heart sound data. Finally, the classification step utilizes appropriate classification algorithms to assign labels to the fragments based on the merged feature set, enabling the accurate classification of different heart sound patterns. By following this four-step process, the proposed method enables robust and comprehensive analysis of heart sound signals, effectively extracting relevant features from both short-term and long-term perspectives. This methodology enhances the overall representation and classification accuracy by incorporating valuable information from the entire heart sound signal.

#### 3.2.1. Pre-Processing

In the pre-processing step, noise reduction, peak detection, segmentation, and fragmentation are performed. The heart sounds in the dataset were recorded in both clinical and non-clinical environments. So, both environmental noise and body organ noise such as lung noise are present in the recordings [37,38]. It is known that fundamental heart sounds and abnormal patterns caused by heart malfunction occur at lower frequencies [39]. Therefore, a low-pass filter was used to eliminate the noise effect in the recordings.

After the noise reduction process, the envelopes of the heart sound signals are extracted to determine the boundaries of the first and second heart sounds. An accepted method for this task is to find all potential peaks using a threshold value and eliminate unnecessary peaks. In our experiments, a methodology called extra peak rejection was used. In this approach, a threshold value equivalent to 30% of the maximum amplitude value is used to determine candidate peaks. Then, to discard extra peaks and identify the correct ones that correspond to the first and second heart sounds, an elimination algorithm is used. Some of the steps in the elimination algorithm are presented below:If two neighboring peaks have a time interval of less than 50 ms, the peak with the smaller amplitude value is rejected, and the other peak is considered a prospective descriptor point.If two neighboring peaks have a time interval of less than 50 ms, the number of heartbeats in healthy individuals should be between 40 and 140 beats per min. Thus, the cardiac cycle, which consists of the systolic and diastolic periods, cannot be shorter than 400 ms or longer than 1500 ms.If two neighboring peaks have a time interval of less than 400 ms and more than 50 ms, the peak with the smaller amplitude value is rejected, and the other peak is considered a prospective descriptor point.If the time interval between the two peaks is more than 1500 ms, it indicates the presence of unidentified peaks. Thus, the threshold value is refined and the rejection steps are applied again.

The last step in pre-processing is fragmentation. The main purpose of fragmentation is to create bigger sample pools, especially for deep learning algorithms. In this work, all recordings were divided into five-second-long windows. The candidate peaks above the threshold line and the extracted S1 and S2 points obtained using the extra peak rejection method are presented in Figure 5.

#### 3.2.2. Feature Extraction

The short-term features contain a total of 27 features, with 14 of them being extracted from the time, high-order statistics, energy, and frequency domains. The rest of the short-term features are extracted from Mel Coefficients. Mel Frequency Cepstral Coefficients (MFFCs) are used in many speech recognition tasks. The MFFC extraction process is inspired by the human cochlea, an organ that vibrates differently depending on the frequency of incoming sounds to facilitate hearing. To mimic this mechanism, first, the power spectrum of each window is computed [31,32]. Then, a narrow filter is utilized to capture the energy near the 0 Hz frequency. Finally, to simulate the human ability to perceive low frequencies, a logarithmic scale is utilized.

Details about the short-term features are presented in Table 1. Regarding the notations in Table 1, each window is denoted as *T* seconds, fs is the sampling frequency, and *X(i)* represents the signal, where *i* varies over the total number of samples. *X(f)* denotes the Fourier transform of the current signal and *Xj(f)* denotes the framed signal, with *j* ranging over the number of frames. *Pj(f)* is the power spectrum of frame *j*. To compute the Mel Frequency Cepstral Coefficients, first, the signal is divided into short frames, and the spectral power density for each frame is calculated. Then, the Mel filter bank is applied to the power spectra, and the energy in each filter is summed. Next, the logarithm of the energies in all filter banks (26 filters are used) is obtained. After taking the logarithm, the Discrete Cosine Transform (DCT) is implemented, and the first 13 coefficients are selected as the MFFC features.

The short-term features have been carefully chosen to capture the essential aspects of the current signal and are highly relevant to the study’s objectives. The mean value of the signal serves as a fundamental measure of the signal’s central tendency, providing insights into its overall magnitude. The median value of the signal complements the mean by capturing the middle value, thus offering a robust estimation less susceptible to outliers. Furthermore, the standard deviation quantifies the dispersion of the signal values around the mean, enabling the assessment of variability within the signal. The mean absolute deviation provides a measure of the average absolute difference between the data points and the mean, giving insights into the overall signal variation. To capture the distributional characteristics, the first quartile (Q1) and third quartile (Q3) reflect the values below which 25 and 75% of the data lie, respectively. The interquartile range, defined as the difference between Q3 and Q1, provides information about the spread of the middle 50% of the data. These quartile-based features are useful for understanding the distributional properties of the signal. In addition, skewness and kurtosis offer insights into the shape of the signal’s distribution. Skewness quantifies the asymmetry of the distribution, whereas kurtosis measures the “peakedness” or “flatness” of the distribution, revealing deviations from a normal distribution. Furthermore, the Shannon entropy and Shannon energy values provide information about the signal’s randomness and spectral characteristics. Shannon entropy assesses the uncertainty or disorder of the signal, whereas Shannon energy, which is computed after applying the fast Fourier transform, captures the spectral entropy. Finally, the Mel Frequency Cepstral Coefficients (MFCCs) (15th to 27th features) are widely used in audio signal analysis for capturing the spectral characteristics of the signal. These coefficients provide information about the signal’s frequency content and have proven effective in various audio classification tasks. By incorporating these specific short-term features, our study aims to comprehensively analyze the current signal, capturing its central tendency, variability, distributional properties, spectral characteristics, and frequency content. The selection of these features is justified by their relevance to the research objectives and their ability to provide valuable insights into the characteristics of the heart sound signals under investigation.

The long-term features consist of six features extracted from the time and high-order statistics domains. Detailed information about the long-term features is presented in Table 2. Regarding the notations in Table 2, it is assumed that there are N occurrences of S1 and S2, and the time indices of those fundamental heart sounds are represented as S1(i) and S2(I), where ‘’i’’ ranges over the total number of S1 and S2, which is N. X(f) denotes the Fourier transform of the current signal.

To determine the information levels of the features, neighborhood component analysis is implemented. According to this analysis, long-term features carry more information than short-term features, and within the short-term features, the MFCCs carry more information compared to the time-, frequency-, and energy-domain features, as shown in Figure 6. In Figure 6, STF indicates short-term features and LTF indicates long-term features. The order of the features in the figure is the same as the order in Table 1 and Table 2.

#### 3.2.3. Feature Pooling

To merge the short- and long-term features, long-term features are added at the end of the short-term features. There are 27 short-term and 6 long-term features. After feature reduction using component analysis, it is understood that all long-term features and 16 short-term features carry a significant amount of information. The potential feature sets and their corresponding dimension sizes are presented in Table 3.

#### 3.2.4. Classification

The PhysioNet dataset has been utilized by numerous machine learning algorithms since becoming publicly accessible. Among these algorithms, traditional machine learning models have proven to be the most popular and successful. To evaluate the effectiveness of various state-of-the-art machine learning techniques and their variant algorithms, we conducted tests on a wide range of models. A comprehensive list of the tested algorithms can be found in Table 4. The fine K-Nearest Neighbor classifier with long short-term features emerged as the most successful among these algorithms.

## 4. Results and Discussion

To assess the classification accuracy of the feature sets shown in Table 3, we employed a range of machine learning techniques, utilizing the 22 classification algorithms presented in Table 4. The abbreviations used in Table 4 are as follows: DT (Decision Tree), NB (Naive Bayes), SVM (Support Vector Machine), KNN (K-Nearest Neighbor), and EM (Ensemble Method).

In all our experiments, the available data were partitioned into two subsets, with 70% of the data reserved for training and the remaining 30% for validation. The computations were performed using MATLAB, and the Machine Learning Toolbox of MATLAB was utilized to extract the performance metrics.

Our analysis revealed that, in comparison to short-term features or long-term features, the use of long short-term features led to an increase in the classification accuracy ranging from 3 to 10% across all employed algorithms, as illustrated in Table 5. These results suggest that long short-term features are better suited for extracting relevant information from heart sound signals and can significantly enhance the performance of classification algorithms.

Table 5 displays the performance metrics of the most accurate algorithms among the tested techniques. It provides a summary of the accuracy results obtained from these techniques. However, for the sake of brevity in this section, the accuracy and other metrics of all 22 algorithms are presented in Table A1, Table A2, Table A3, Table A4, Table A5 in Appendix A. Analyzing the results presented in these tables, we can see that using the proposed technique can increase accuracy by more than 10%, especially in cases where the classification accuracy is very low, e.g., 70% or lower. Another important finding is that the Selected Merged Features (SLSTF) achieved better classification accuracy than either the short-term or long-term features alone. Our experiments demonstrated that incorporating long-term information into window-specific short-term features can significantly enhance classification accuracy.

An additional observation that can be derived from Table 5 is that the accuracies of the K-Nearest Neighbor and Ensemble Method algorithms remained consistently close across different feature sets. The findings of this study indicate that the K-Nearest Neighbor (KNN) and Ensemble Method algorithms exhibited remarkable similarity in terms of accuracy across various feature sets. KNN is a straightforward and intuitive classification algorithm that assigns labels to data points based on their proximity to labeled instances. It is not bound by any assumptions regarding the underlying data distribution and demonstrates versatility in accommodating different feature sets. By calculating distances between data points and identifying the nearest neighbors, KNN classifies new data points based on the labels of these neighbors. The relatively close results in terms of accuracy obtained for different feature sets in KNN can be attributed to the algorithm’s heavy reliance on the distance metric. When distinct feature sets yield comparable distances, it results in similar classification accuracy. Ensemble Methods are characterized by their amalgamation of multiple learning algorithms to enhance predictive performance. Functioning as a collective unit, Ensemble Methods combine individual models to arrive at a final prediction. The incorporation of weak learners, which are individual models with moderate accuracy, allows Ensemble Methods to generate a strong learner by leveraging the diverse knowledge encapsulated within the constituent models. This diversity facilitates the capture of varied aspects within the data, ultimately contributing to improved accuracy. Notably, Ensemble Methods exhibit closely aligned accuracy levels for different feature sets owing to their ability to harness the collective expertise of the constituent models, rendering them less susceptible to the idiosyncrasies of individual feature sets. The aforementioned observations highlight the robust decision-making processes inherent in KNN and Ensemble Method algorithms, which enable these techniques to maintain consistent accuracy across diverse feature sets. These characteristics make them reliable choices for addressing classification tasks in scenarios involving varying feature representations.

For optimal performance, the highest accuracy is achieved by employing specific algorithms within their respective techniques. The most accurate algorithm among the Decision Tree models was the Fine Tree algorithm. Among the Naive Bayes models, the Gaussian algorithm achieved the best results. The Fine Gaussian algorithm was found to be the most accurate among the Support Vector Machine models. In the case of the K-Nearest Neighbor models, the Weighted KNN algorithm demonstrated superior accuracy. Finally, among the Ensemble Methods, the Subspace KNN algorithm was found to be the most accurate.

The proposed technique raises an important discussion point regarding whether the windows used for classification should be isolated from neighboring information. To address this issue, it is necessary to consider whether a normal record can contain abnormal fragments, and vice versa. It is reasonable to expect that if a heart sound record is normal, all fragments of that record will be classified as normal, and vice versa. It is also assumed that each fragment possesses sufficient representative properties for the record it belongs to. However, this assumption may not always hold, as some fragments may not contain enough origin information to be accurately classified. One possible solution to this issue is to incorporate origin information into the fragments. The results presented in the tables indicate that adding origin information through long-term features can significantly improve the robustness of nearly all classification algorithms.

## 5. Conclusions

Phonocardiogram recordings carry a lot of useful information about the heart condition, and machine learning algorithms can be employed for automatic diagnosis purposes. Thus, a variety of methods have been proposed for automatic heart sound classification. This work aims to enhance classification accuracy by changing the perspective in the feature extraction process. Generally, heart sounds are divided into small fragments, and classification is carried out on those fragments. However, this approach lacks overall information about the heart sound instance and its neighboring fragments. The proposed feature extraction method is designed to compensate for this deficiency by incorporating two different time scales: the first corresponds to the individual fragment, and the second encompasses the entire heart sound instance, which consists of the respective fragment. Features are extracted from both time scales, and with the help of both feature sets, classification accuracy and other performance metrics are improved across various state-of-art classification algorithms.

By employing a two-time-scale approach that incorporates both short-term and long-term features, we aimed to address the limitations of existing methods that primarily rely on analyzing small fragments of heart sounds. Through comprehensive feature extraction from both the respective fragment and the entire heart sound instance, our proposed method captures a more holistic representation of the heart sound signal. This approach allows us to leverage the valuable information carried by the long-term features, providing a more accurate and robust analysis. A comparative analysis of our proposed method with state-of-the-art classification algorithms showcases its superiority in terms of classification accuracy and other performance metrics. The incorporation of both time scales leads to a significant improvement in the classification results, demonstrating the importance of considering the overall information of the heart sound instance and its neighboring fragments. The findings of this study have important implications for the field of automatic heart sound classification. Our proposed method enhances not only classification accuracy but also our understanding of the underlying patterns and characteristics of heart sounds. This, in turn, contributes to the advancement of computer-aided diagnosis systems for cardiovascular diseases.

Further research directions may include exploring additional feature extraction techniques and refining the classification algorithms to achieve even higher accuracy. Additionally, the generalization and applicability of the proposed method can be evaluated on larger and more diverse datasets to ensure its robustness across different patient populations and clinical settings.

In summary, this study highlights the significance of incorporating both short-term and long-term features in the feature extraction process for automatic heart sound classification. The proposed method offers improved accuracy and provides valuable insights into heart sound analysis. With continued advancements in machine learning and signal processing techniques, the field of automatic heart sound classification holds great promise for improving cardiovascular disease diagnosis and patient care.

## Figures and Tables

**Figure 1 sensors-23-05835-f001:**
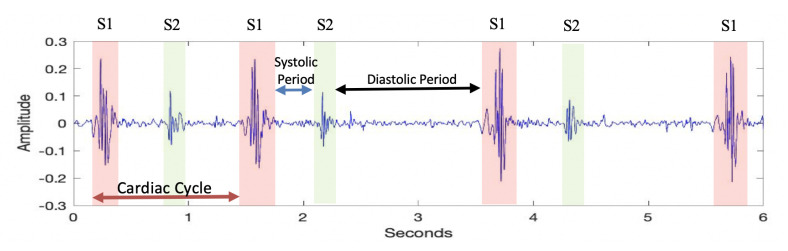
Fundamental heart sounds, cardiac cycle, and systolic and diastolic periods.

**Figure 2 sensors-23-05835-f002:**
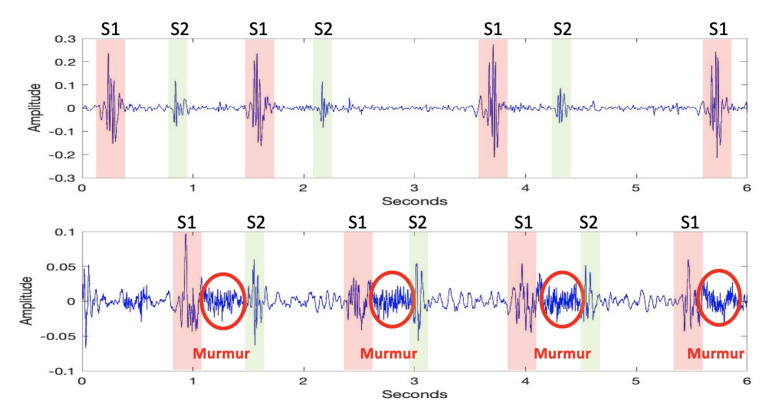
Heart sound graph of a healthy individual (**upper**). Heart sound graph of an unhealthy individual (**lower**).

**Figure 3 sensors-23-05835-f003:**
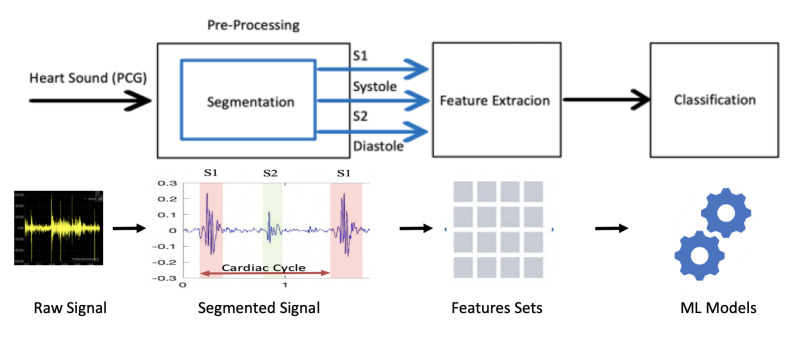
The classical method of heart sound classification.

**Figure 4 sensors-23-05835-f004:**
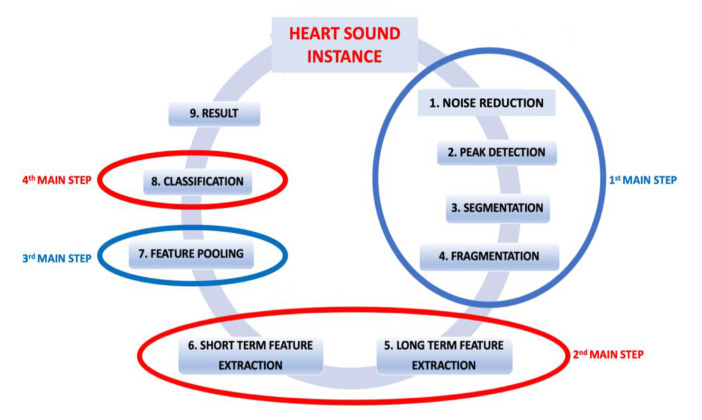
Flowchart of the proposed method.

**Figure 5 sensors-23-05835-f005:**
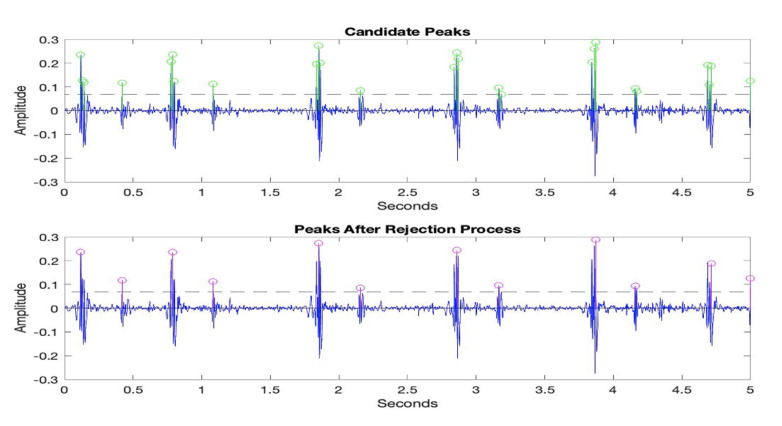
Peaks with values above the threshold (**upper**). Peaks after extra peak rejection process (**lower**).

**Figure 6 sensors-23-05835-f006:**
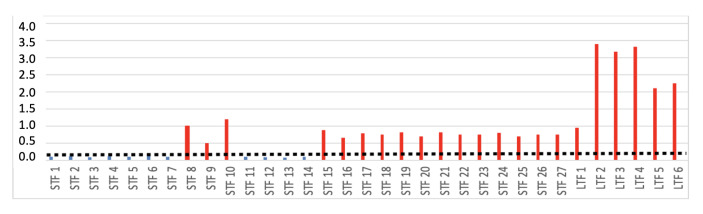
Information levels of the features. Dotted line is used to display the threshold level. Blue squares are the ones below threshold, and red lines are the ones above threshold.

**Table 1 sensors-23-05835-t001:** Detailed profile of the short-term features.

Feature	Mathematical Description	Explanation
1	$\frac{\left(\sum_{i=1}^{Tfs}X(i)\right)}{Tfs}$	The mean value of the current signal.
2	$\frac{\left(X((Tfs)/2))+X((Tfs)/2)+1\right)}{Tfs}$	The median value of the current signal.
3	$\sqrt{\frac{\sum_{i=1}^{Tfs}{\left((X(i))-Mean\right)}^{2}}{Tfs}}$	The standard deviation of the current signal.
4	$\frac{\sum_{i=1}^{Tfs}\left|(X(i))-Mean\right|}{Tfs}$	The mean absolute deviation.
5	$\frac{1}{4}\left(n+1\right)$th term	first quartile of the current signal (Q1).
6	$\frac{3}{4}\left(n+1\right)$th term	third quartile of the current signal (Q3).
7	Q3–Q1	The interquartile range, which is the difference between the first and third quartiles.
8	$\sum_{i=1}^{N}{\left((X(i)-m)/s\right)}^{3}.\frac{1}{N}$	The skewness of the current signal. X(i) is an individual score; “*m*” is the population mean; “*s*” is the population standard deviation; *N* is the population size.
9	$\sum_{i=1}^{N}{\left((X(i)-m)/s\right)}^{4}.\frac{1}{N}$	Kurtosis of the current signal. X(i) is an individual score; “*m*” is the population mean; “*s*” is the population standard deviation; *N* is the population size.
10	$\sum_{i=0}^{Tfs}X(i)\log(X(i))$	Shannon entropy value of the current signal.
11	$\sum_{f=0}^{N}X(f)\log(X(f))$	After applying the fast Fourier transform, the Shannon energy is computed (spectral entropy of the current signal). Note: X(f) denotes the Fourier transform of the current signal.
12	$Max(\sum_{i=0}^{N-1}X\left(i\right)e^{-\frac{i2\pi kn}{N}})$	The maximum frequency (Hz) after applying the fast Fourier transform.
13	$Max(\sum_{i=0}^{N-1}X\left(i\right)e^{-\frac{i2\pi kn}{N}})$	The maximum frequency spectrum value after applying the fast Fourier transform.
14	$Energy[Max\left(\sum_{i=0}^{N-1}X\left(i\right)e^{-\frac{i2\pi kn}{N}}\right)]$	The ratio of the energy of the maximum frequency to the total energy.
5th to 27th	13 coefficients from the MFFCs	Mel Frequency Cepstral Coefficients

**Table 2 sensors-23-05835-t002:** Detailed profile of the long-term features.

Feature	Mathematical Description	Explanation
1	$T12=\frac{abs(\sum_{i=1}^{N}S1(i)-S2(i))}{N}$	This feature is used to represent the mean value of the systolic intervals.
2	$T21=\frac{abs(\sum_{i=1}^{N}S2(i)-S1(i+1))}{N-1}$	This feature is used to represent the mean value of the diastolic intervals.
3	$SD12=\sqrt{\frac{\sum_{i=1}^{N}{\left((S1(i)-S2(i))-T12(a)\right)}^{2}}{N}}$	This feature is used to represent the standard deviation of the systolic intervals.
4	$SD21=\sqrt{\frac{\sum_{i=1}^{N}{\left((S2(i)-S1(i+1))-T21(a)\right)}^{2}}{N-1}}$	This feature is used to represent the standard deviation of the systolic intervals.
5	$\frac{\begin{array}{c} Total\;Number\;of\;Peaks\\ After\;Extra\;Peak\;Rejection \end{array}}{Total\;Number\;of\;Peaks}$	Ratio of rejected peaks to total peaks.
6	$\frac{\begin{array}{c} Mean\;Amplitude\;Value\;of\;the\;Peaks\\ After\;Extra\;Peak\;Rejection \end{array}}{Mean\;Amplitude\;Value\;of\;All\;Peaks}$	Amplitude value of rejected peaks to all.

**Table 3 sensors-23-05835-t003:** Feature sets.

Abbreviation	Number of Features	Explanation
STF	27	Short-term features.
LST	6	Long-term features.
LSTF	33	Short-term features + long-term features.
SSTF	16	Short-term features after feature reduction.
SLSTF	22	Short-term features after feature reduction + long-term features.

**Table 4 sensors-23-05835-t004:** Classification techniques and algorithms.

Technique	Algorithms					
	1	2	3	4	5	6
Decision Trees	Fine Tree	Medium Tree	Coarse Tree			
Naive Bayes (NB)	Gaussian NB	Kernel NB				
Support Vector Machines (SVMs)	Linear SVM	Quadratic SVM	Cubic SVM	Fine SVM	Medium SVM	Coarse SVM
K-Nearest Neighbors (KNNs)	Fine KNN	Medium KNN	Coarse KNN	Cosine KNN	Cubic KNN	Weighted KNN
Ensemble Methods	Boosted Tree	Bagged Tree	Subspace Discriminant	Subspace KNN	RUSBoosted Tree	

**Table 5 sensors-23-05835-t005:** Classification accuracy of the tested algorithms with different feature sets.

Technique	Feature Sets				
	STF	LTF	LSTF	SSTF	SLSTF
Decision Trees	78.6%	77.9%	**87.8%**	83%	70%
Naive Bayes	68%	75.2%	**83.3%**	71.6%	76%
Support Vector Machines	81.6%	74.5%	**91.2%**	79%	80.5%
K-Nearest Neighbors	87.3%	86%	**89.9%**	86.9%	88%
Ensemble Methods	90.3%	89%	**92.7%**	90.5%	90.1%

## Data Availability

The raw dataset, which was downloaded from the PhysioNet website, is available at https://physionet.org/content/challenge-2016/1.0.0/ (accessed on 29 April 2023).

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
