# Peer review of "A New Method for Heart Disease Detection: Long Short-Term Feature Extraction from Heart Sound Data"

_sensors, 2023, doi:10.3390/s23135835_

Round 1
Reviewer 1 Report
In the article “a new method for heart disease detection: the proposed long-short-term feature extraction on heart sound data” authors have decided this study proposes a novel approach that integrates window-based features extracted from the entire signal, thereby improving the overall accuracy of traditional machine learning algorithms. The manuscript is written well but the discussion is quite short. The discussion can be extended by comparing results from this method to the existing one in the literature and writing down the strengths and shortcomings of this method.
(i) Why feature sets are quite close in K-Nearest Neighbor and Ensemble Methods but not in other three techniques? Explain in detail?
(ii) What is the best combination of algorithms in classification techniques to get optimum results?
(iii) Keep the space between lines in Table 4.
Author Response
Thank you very much for your valuable comments and recommendations. I tried to respond to all the comments and I believe with those additions the paper will look more sophisticated.

Reviewer 2 Report
1. The manuscript aims at presenting an study using using LSTM approach. But the title of the paper itself seems to be repetitive in sense "new method" and "proposed method". Clarity required in title of the paper.
2. Organization of the paper needs improvement in presenting the paper with basics of the proposed approach is not found in the paper. Only glimpse of the proposed title is presented.
3. Figures presented in the paper needs refinement in terms of quality of the paper. (E.g Fig.2, Fig. 3 etc)
4. Psuedo procedure should be presented in the paper for better clarity on the paper.
5. The study presented with comparison of several short-term features needs improvement in terms of explanation. The logic of introducing those in the context is necessary? Can the authors clarify on it?
6. Results presented in Tables A.1 to A.5 needs clarity. Whether the results obtained should be presented in terms of floating values (e.g 78.6%). The authors presented the values as (62,3%). what is the value representation in the paper. There is clarity required in this.
7. The conclusion arrived out of the study is not convincing in terms of research outcomes. Rather, authors simply mentioned as the "values are low". It could not be accepted.
8. Overall the paper needs improvements in several aspects as illustrated above.
9. References presented in the paper should follow same style of representation. It is not of common format (E.g Ref 36, 39) doesnt follow standard approach.
The paper could be proof read in a few areas, grammar refinements could be done in certain areas.
Author Response
Thank you very much for your valuable and to-the-point comments/recommendations. I tried to respond to all. Finally, I believe that with these additions and checks, the manuscript becomes more sophisticated. I look forward to good news from you.
Please kindly refers to the response to Reviewer 2.
Very respectfully, Mesut G.(Ph.D.)

Round 2
Reviewer 2 Report
The authors have done major revision considering the comments suggested. The paper in its present form is insightful and seems to be a new version as compared to older one. The changes done in the paper is quiet compromising and accepted.
The paper could be considered for publication in its present form.